# Characterization of Pure and Blended Pellets Made from Norway Spruce and Pea Starch: A Comparative Study of Bonding Mechanism Relevant to Quality

**Anthony Ike Anukam** [1,*], **Jonas Berghel** [1], **Stefan Frodeson** [1], **Elizabeth Bosede Famewo** [2] **and Pardon Nyamukamba** [3]

1 Environmental and Energy Systems, Department of Engineering and Chemical Sciences, Karlstad University, 651 88 Karlstad, Sweden; jonas.berghel@kau.se (J.B.); stefan.frodeson@kau.se (S.F.)

2 Electron Microscopy Unit, Central Analytical Laboratory, Faculty of Science and Agriculture, University of Fort Hare, Alice 5700, South Africa; ritayomidef@gmail.com

3 Technology Station Clothing and Textiles, Cape Peninsula University of Technology, Symphony Way, Bellville 7580, South Africa; nyamukambap@cput.ac.za

* Correspondence: anthony.anukam@kau.se; Tel.: +46-54-700-1664

**Abstract:** The mechanism of bonding in biomass pellets is such a complex event to comprehend, as the nature of the bonds formed between combining particles and their relevance to pellet quality are not completely understood. In this study, pure and blended biomass pellets made from Norway spruce and pea starch were characterized using advanced analytical instruments able to provide information beyond what is visible to the human eye, with intent to investigate differences in bonding mechanism relevant to quality. The results, which were comprehensively interpreted from a structural chemistry perspective, indicated that, at a molecular level, the major disparity in bonding mechanism between particles of the pellets and the quality of the pellets, defined in terms of strength and burning efficiency, were determined by variation in the concentration of polar functional groups emanating from the major organic and elemental components of the pellets, as well as the strength of the bonds between atoms of these groups. Microscopic-level analysis, which did not provide any clear morphological features that could be linked to incongruity in quality, showed fracture surfaces of the pellets and patterns of surface roughness, as well as the mode of interconnectivity of particles, which were evidence of the production of pellets with dissimilarities in particle bonding mechanism and visual appearance.

**Keywords:** pelleting; functional groups; biomass; pellet strength; combustion efficiency

## 1. Introduction

The use of biomass for the production of energy and valuable chemicals is gaining attention because it is renewable, clean, cheap, and readily available. There are several types of biomass (such as wood, starch, agricultural residues, energy crops, and industrial and municipal solid wastes) suitable for energy conversion after subjecting the biomass to pretreatment processes like pelleting, and the pretreatment method is often determined by the conversion pathway intended for the biomass; however, the need to press biomass into pellets arises from its heterogeneous nature and low energy density, which makes its conversion in energy production systems very problematic [1]. This means that pelleting of biomass is basically a technique used to improve biomass characteristics in the form of a pellet with regular shape, along with higher density, strength, and durability, as well as excellent combustion characteristics and low ash content, which are factors used to define good-quality biomass pellets [2,3]. The improved quality of pelletized biomass makes it suitable as a fuel for use in household

heating boiler systems, cooking, and electricity generation. Most biomass pellets are usually made from wood such as Norway spruce; the global fuel pellet market experienced expeditious growth in recent times, and it is anticipated to have even faster growth in the near future [4]. Even though almost all biomass can be pelletized, not all are likely to form durable pellets because of variations in characteristics; hence, different types of biomass are sometimes blended for the purpose of improving quality. Additionally, because of issues related to dust formation and self-ignition, additives such as starch may be added to the biomass during pelleting in order to increase the overall quality of the pellets [2,5]. However, the production of durable biomass pellets is always challenged by a host of factors, including a lack of fundamental understanding of the bonding mechanism of major components during pelleting, type of materials to be blended with the biomass, how these materials affect the mechanism of bonding, and different pellet quality parameters. Other factors impacting the production of durable biomass pellets include types of biomass for pelleting, moisture content of the biomass, organic and elemental constituents of the biomass, particle size and distribution, and pellet press compression force and temperature [6–9]. Most of these factors were studied by other researchers, yet differences in the mechanism of bonding between pure and blended biomass pellets relevant to quality still need to be investigated. As previously mentioned, wood is a type of biomass most commonly used in the production of fuel pellets and remains a major source of energy in most countries. On the other hand, starch is perceived as a good additive to wood for the production of durable biomass pellets. However, to the best of the authors' knowledge, the production of biomass pellets from pure starch and other materials (such as wood) blended with starch, with the aim to investigate the mechanism of bonding relevant to quality, is sparsely studied. To lay the groundwork for a better understanding of what was investigated in this study, the section below presents a brief synopsis of the chemistry of wood and starch.

## 1.1. Overview of the Chemistry of Wood and Starch

It was reported that wood such as Norway spruce remains the most common material for the production of fuel pellets in Sweden, and that starch has excellent applications in a handful of industrial sectors such as biofuel, food, pulp, and paper [5,10]. The properties of these two materials are controlled by complex interactions between their physical and chemical structure. For instance, the structural characteristics of wood are such that its cells are made up of varying proportions of three major substances (cellulose, hemicellulose, and lignin), whose structures are bound by functional groups that are responsible for the behavior of wood in many conversion processes. Depending on the type of wood, the weight percentages of cellulose, hemicellulose, and lignin in wood range from 40% and 50% for cellulose, from 20% to 28% for hemicellulose, and from 25% to 30% for lignin [3,7,11,12]. The wood structure is complex and anisotropic with optimized hierarchical levels that span from the macro to the micro, molecular, and even nano scale. The structure of the cellulose component of wood is linked by β-(1,4)-glycosidic bonds with a high degree of polymerization, while that of hemicellulose is partially substituted by acetyl groups with a lower degree of polymerization in comparison to cellulose. The substituted acetyl groups in the hemicellulose structure means that the hydroxyl groups (–OH) at carbon positions $C_2$ and $C_3$ are partly substituted by *O*-acetyl groups, one of the adhesive degradation products of hemicellulose responsible for natural bonding [13,14]. Lignin is complex in nature and contains aromatic rings that are responsible for its glue effects; the thermosetting properties of lignin are exhibited at temperatures above or equal to 100 °C, and the adhesive nature of thermally melted lignin significantly contributes to the strength and durability of pellets made from lignocellulosic biomass [15–17].

Starch, on the other hand, is a soft, white, and tasteless powder whose usefulness as an additive in improving bonding properties of biomass pellets cannot be overemphasized. This is because of the chemical structure of its two major monomer units known as amylose and amylopectin [18]. Just like wood, the structures of these two monomer units of starch are also bound by functional groups that confer specific properties. The functionality of starch depends on these groups and the

average molecular weight of its amylose and amylopectin contents [19]. Both of these constituents of starch consist of chains of α-(1,4)-glycosidic bonds that are linked by ᴅ-glucose residues connected via α-(1,6)-glycosidic linkages, thereby forming polymer branches [19,20]. More often than not, the relative weight percentages of amylose and amylopectin in starch range from 18% to 33% for amylose, and from 72% to 82% for amylopectin [19]. Morphological features of starch include perfectly spherical particles with many void spaces exhibiting both internal and external surface areas that are determined by the shape and size of its particles, which induces higher molecular mobility when starch is heated (gelatinization) [19,21].

The structures of the major components of both wood and starch can be found in References [12,22].

### 1.2. Purpose of Study

Several studies reported the pelleting of different types of biomass and their blends, with most of the studies focusing on material and process parameters including the mechanism of bonding from different viewpoints [2,23–27]. Despite the numerous studies, the exact mechanism involved in bonding in biomass pellets and the source of inter-particle bonding relevant to pellet quality are not fully understood and still a subject of debate, particularly when the pellets are made from a blend of two or more different biomass materials. Not much was undertaken, in terms of research, to study and establish, from a structural chemistry perspective, differences in the mechanism of bonding between pure and blended biomass pellets relevant to the production of good-quality pellets. Investigations of this sort require a systematic approach that takes into account the use of state-of-the-art analytical techniques that can go beyond standard tests and visual assessments. The techniques must be able to provide information beyond what can be visualized by the human eye and offer evidence of quality without having to rely on special quality assessment tools. Therefore, this study aims to investigate and establish differences in the mechanism of bonding, relevant to quality, between pure and blended pellets made from Norway spruce and pea starch, relying on molecular, microscopic, and thermal analyses provided through the use of advanced analytical instruments able to offer information beyond what can be visualized by the naked eye.

The choice of the materials used in the study, the reason for blending, and the blend ratio are explicitly described in the section below.

## 2. Materials and Methods

### 2.1. Sample Preparation Steps

While Norway spruce was selected for this study because of its availability and suitability as a feedstock for heat and power production, pea starch was chosen for its cost-effectiveness and suitability as an additive in pelleting processes. Whereas the pea starch was an industrial secondary starch obtained in powder form from Lyckeby, Kristianstad, Sweden, Norway spruce was obtained in the form of logs from a local sawmill in Karlstad, Sweden and was cut into chips. The two raw materials, upon procurement in their pure form, were placed in small plastic bags and stored at room temperature. Prior to pelleting, the materials were dried in an oven at 50 °C for 24 h. Because starch was obtained in powder form, it did not require milling, but was sieved to a particle size of 0.3 mm, according to American Society for Testing and Materials (ASTM) standards (D 2013-72). Norway spruce was milled to a particle size of <1 mm using a Wiley mill and also sieved to obtain a final particle size equal to that of pea starch. Appropriate quantities of each sample were separately weighed and mixed with water to obtain a final moisture content of 10% (wet basis). This was because moisture is often required during pelleting to reduce friction and to avoid blocking of the pellet press die; under these circumstances, however, adding the right amount of water is very vital, as material moisture content must not exceed 10% for pellets made from woody biomass, and a maximum moisture content of 15% for pellets made from grass [6,28]. Preceding these steps and soon after procurement, milling, and sieving, the initial moisture contents of the two samples (Norway spruce and pea starch) were determined according to

standards [29]. The determination of moisture content was done on a wet basis by weighing the two samples and placing them in an oven at 105 °C for 24 h and weighing again to determine final weight.

### 2.2. Preparation of Blend

A 50/50 blend ratio of milled Norway spruce and pea starch was prepared by weighing 45 g of each respective material in order to achieve a uniform weight ratio for blending. Both materials were placed in a plastic container and thoroughly mixed together for 10 min using a 220-V Janke and Kunkel Ika-Werk mixer rotated at a speed of 60 rpm. The blend was prepared by mixing with water to achieve a moisture content (10%) equal to that of the pure samples described in the previous section. The procedure for achieving the desired moisture content for the blend was such that an appropriate amount of water was added to 90 g of the mixture and then mechanically rotated again for another 10 min. Shortly after rotation, the mixture was sealed in small plastic bags and stored for 48 h to allow the moisture to settle in and to prevent further absorption of moisture from the surroundings. Care was taken during the preparation of mixtures with water so as to avoid the formation of rigid irreversible gel by pea starch, which may alter the analysis results. The 50/50 percentage blend was a random selection aimed at keeping the ratio of the admixture of Norway spruce and pea starch equal for a fair comparison of the mechanism of bonding between the pure and blended pellet samples made from the two materials. The pellet production process of the pure and blended samples of Norway spruce and pea starch is described in the section below.

### 2.3. The Pellet Production Process

Pure and blended pellets were produced from Norway spruce and pea starch using a single pellet press available at the Department of Engineering and Chemical Sciences of Karlstad University in Sweden. Figure 1 presents the laboratory-scale pellet press set-up used in the pellet production process.

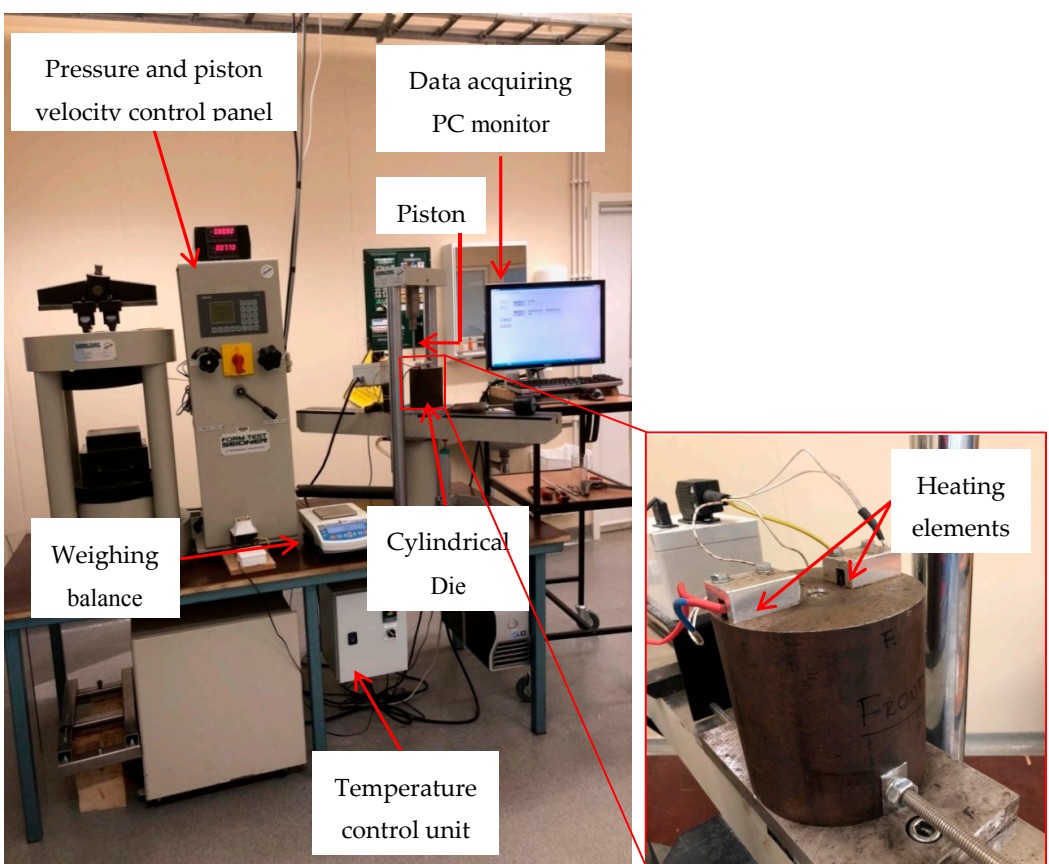

**Figure 1.** The single-unit pellet press set-up used in this study with labeled parts.

The press consisted of a cylindrical die made of hard steel, about 8 mm in diameter, with connected heating elements and thermal insulation. A temperature control unit was used to regulate die temperature from ambient conditions to a maximum temperature of 100 °C. The bottom end of the cylindrical die was closed with a removable plastic piston stopper. Compression force was mechanically applied with a piston pointing directly toward the top opening of the die filled with exactly 1 g each of the pure and blended samples of milled Norway spruce and pea starch. Piston movement was controlled by the entire pellet press system, and pressure was applied to the piston through a load cell whose maximum compression force was about 2000 kN. Before each use and when changing the raw materials, the die was rinsed several times with acetone and cleanly wiped with a paper towel. Each of the pure and blended samples was compressed at a velocity of 30 mm·min$^{-1}$ until a maximum pressure of 16 kN was reached. The holding time for every test was set at 10 s, which is in agreement with studies conducted by Nielsen et al. [30] and by Nguyen et al. [31]. The pellets were removed from the die by removing the stopper and mechanically pushing out the pellets, slowly, at a velocity of 5 mm·min$^{-1}$. According to a previous study [7], the optimum pelleting conditions under which the pure and blended samples of Norway spruce pellet (NSP) and pea starch pellet (PSP) were produced at a laboratory scale are shown in Table 1.

**Table 1.** Optimum pelleting conditions under which pure and blended samples of Norway spruce pellet (NSP) and pea starch pellet (PSP) were produced [7].

| Parameter | Pelleting Conditions |
| --- | --- |
| Moisture content | 10% (prior to pelleting) |
| Die temperature | 100 °C |
| Compression force | 16 kN |
| Holding time | 10 s |
| Piston velocity | 30 mm·min$^{-1}$ (compression phase) |
| Push out velocity | 5 mm·min$^{-1}$ (friction phase) |
| Raw material quantities | 1 g (each of milled material) |

*2.4. Analytical Techniques*

It is worth mentioning that, in addition to the data generated on differences in the bonding mechanism of the pure and blended samples of NSP and PSP used in this study, the quality of the pellets was interpreted in terms of strength and combustion efficiency. The study did not incorporate the use of special tools that are restricted to the determination of the quality parameters; instead, the study relied on evidence of the above quality parameters from the data obtained through the use of state-of-the-art molecular, microscopic, and thermal analytical instruments. Therefore, the subsections below present specific information about the analytical instruments used in this study and describe the experimental procedures related to how the instruments were used. For the sake of reproducibility, all analyses presented in this study were repeated in triplicate, and average values were reported.

2.4.1. Content Analysis

Assessing biomass for any conversion or pretreatment process often requires a good understanding of its basic composition and characteristics [1,3,32]. Thus, the major components of the pure and blended pellet samples believed to have greater influence on bonding and pellet quality were considered during the content analysis. Content analysis of both pure and blended samples of NSP and PSP was conducted to determine major organic and elemental components in terms of percentage composition. Details of how these components were quantified are given in Section 2.4.3. However, subsequent analytical instruments were used to investigate and establish differences in the mechanism of bonding between the pure and blended pellet samples relevant to bonding and quality of the pellets.

The major organic components of PSP (100%) were quantified by a differential scanning calorimeter (DSC), whose make, model, and principle of operation are detailed in Section 2.4.4.

For the major elemental constituents of the pure and blended pellet samples, an energy-dispersive X-ray spectroscope (EDX) connected to a scanning electron microscope (SEM) was used. The analysis (EDX) involved the generation of characteristic X-rays that revealed elements present in the pellet samples. The procedure was such that, the SEM, whose model and mode of operation are detailed in Section 2.4.5, created an image of the samples by scanning them with an electron beam. During this process, an X-ray was emitted from the samples, which was then collected by the EDX instrument after bombardment by an electron beam. The collected X-rays were characteristic of the atomic structure of the elements contained in the pellet samples. Energy level was used to sort the X-rays, after which a plot of X-ray energy versus frequency was obtained. This plot gave an indication of the elements present in the pellet samples and the percentage composition of each element.

The results obtained from this analysis (content analysis) laid the groundwork for data interpretation from subsequent analysis involving the use of other advanced analytical instruments.

### 2.4.2. Fourier-Transform Infrared Spectroscopic Analysis (FT-IR)

Since bonding, from a chemistry perspective, relates to perpetual attraction between atoms, ions, or molecules, there is a need to firstly identify the type of forces acting between individual particles of the pure and blended samples of NSP and PSP prior to identifying chemical alterations caused by blending. However, the goal of this analysis was to determine the influence of chemical alteration on bonding and establish differences in the mechanism of bonding between the pure and blended pellet samples relevant to the quality of the pellets. Analysis of this nature would require molecular-level identification of key functional groups with significant impact in bonding. Therefore, FT-IR analysis was invaluable in helping to achieve this goal. The procedure for the FT-IR analysis is briefly described below.

FT-IR measurements of pure and blended samples of NSP and PSP were carried out using a Perkin Elmer FT-IR spectrometer (Spectrum Two). Spectra were scanned at a resolution of 4 cm$^{-1}$ with 32 scans collected in the spectral range of 4 000 cm$^{-1}$ to 500 cm$^{-1}$.

### 2.4.3. Thermogravimetric Analysis (TGA)

Combusting biomass at elevated temperatures creates chemical modifications that affect the properties and performance of the biomass in any thermal pretreatment or conversion process, and the extent of chemical modification depends on the temperature level and the duration of conditions of thermal exposure [33,34]. In thermal analysis of biomass materials, physical and chemical changes associated with the material are usually determined as a function of temperature. TGA analysis of pure and blended samples of NSP and PSP was conducted for the following reasons: (1) to determine and quantify major organic constituents based on the degradation temperature of each component; (2) to determine differences in the temperature range of chemical modifications relevant to bonding and pellet quality; (3) to determine the combustion characteristics/efficiency of the pellets relevant to quality. Since the pellet samples were made from two different materials (Norway spruce and pea starch), TGA analysis was undertaken to specifically establish how, if any, differences in modification temperature can facilitate particle bonding. A plot of weight loss against temperature is usually obtained from TGA analysis and represents the thermal behavior of a sample and the media through which components of the sample can be quantified based on the degradation temperature of each component [7,35]. Therefore, the major organic components of NSP (100%) and the NSP/PSP (50%/50%) blend were quantified from the plot obtained from this analysis, done in accordance with (after moisture evaporation) differences in the thermal decomposition temperatures of each component, as the pellet samples lost weight as temperature increased. The major organic components of biomass are commonly differentiated and identified using TGA due to differences in the structure of the components [33,36–38]. The procedure for TGA analysis is described below.

Thermal analyses of pure and blended samples of NSP and PSP were performed with a Perkin Elmer TGA 4000 instrument in which samples were firstly placed in the combustion chamber of the

instrument and ignited via software programming. The samples were heated from about 28 to 760 °C at a heating rate of 20 °C·min$^{-1}$ under a nitrogen gas flow rate of 19.8 mL·min$^{-1}$ in order to maintain a non-reactive atmosphere as the analysis progressed. The weight loss of the samples was typically monitored as the samples were heated, cooled, and isothermally held.

### 2.4.4. Differential Scanning Calorimetric Analysis (DSC)

According to Back [39], sufficient bonding areas are activated when biomass components are plasticized above their transition temperatures (T$_g$). The comparative assessment of combustion profiles helps to determine the series of stages that characterize the thermal behavior of biomass [40]. DSC analysis was used as a complementary technique to TGA to determine the T$_g$ or softening temperature of the pellet samples as a function of heat flow, as well as establish phase changes caused by thermal exposure as result of blending. The intention was to compare the T$_g$ of the pure and blended pellet samples and determine if this would have an impact on how bonding occurs in pelleting of different biomass materials. Polymer viscosity significantly deceases to show obvious flow characteristics that are relevant to bonding at T$_g$ higher than 50 °C [7,41]. As previously mentioned, the major organic and elemental components of PSP (100%) used in this study were determined by DSC according to the method described by Moorthy et al. [42].

A Perkin Elmer DSC 6000 was used to determine T$_g$ caused by thermal degradation at specific heat flow during thermal analysis of pure and blended samples of NSP and PSP. The samples were heated to a temperature of approximately 440 °C at a heating and nitrogen gas flow rates similar to those given for TGA analysis. A computer was used to monitor the temperature and regulate the rate at which temperature changed for a given amount of heat.

### 2.4.5. Scanning Electron Microscopic Analysis (SEM)

To understand sample composition and morphological characteristics relevant to bonding at a micro scale, as well as view structural changes that are vital factors of pellets quality in terms of strength, a high-resolution SEM that offers surface morphology of samples at high magnifications was used. The intention was to use images generated by the SEM to determine visible differences in particle bonding mechanism between the pure and blended pellet samples that were relevant to quality through the identification of quality features from the images.

The morphological characterization of pure and blended samples of NSP and PSP was conducted with a JEOL (JSM-6390LV) model SEM instrument fitted with an EDX analyzer, which was used for the analysis of major elemental components. Samples were placed on an aluminum holder stub using a double-sided sticky carbon tape and sputter-coated with gold (Au) using an Eiko IB3 Ion Coater. Sputter-coating was necessary to increase conductivity and prevent charging effects that may lead to blurred images, which may hamper detailed interpretation of results. Thereafter, samples were mounted on stub holders in the sample chamber of the SEM for morphological examination.

## 3. Results

### 3.1. Compositional Analysis

The results obtained from this entire study were comprehensively interpreted from a structural chemistry viewpoint and were also used to predict the relevance of bonding to the quality of the pellets. The presented data from the analytical instruments used in the study were substantiated using information from the existing literature.

As previously mentioned, compositional analysis was undertaken to determine major organic and elemental components as a first step to the study. Table 2 presents the percentage composition of the major organic and elemental components of pure and blended samples of NSP and PSP obtained from the content analysis test presented in Section 2.4. The percentage error for the reported values

was within ±0.3 and 1.0% for the major organic constituents, and within ±0.7 and 1.2% for the main elemental components.

**Table 2.** Percentage composition of major organic and elemental components of pure and blended samples of NSP and PSP. ND—not determined.

| | NSP (100%) | PSP (100%) | NSP/PSP (50%/50%) |
|---|---|---|---|
| **Major Organic Constituents (%)** | | | |
| Cellulose | 41.6 | ND | 21.7 |
| Hemicellulos | 23.2 | ND | 12.3 |
| Lignin | 32.9 | ND | 16.5 |
| Amylose | ND | 28.2 | 19.2 |
| Amylopectin | ND | 70.3 | 27.8 |
| **Major Elemental Components (%)** | | | |
| C | 46.2 | 33.4 | 40.8 |
| H | 6.4 | 5.8 | 5.2 |
| O | 44.6 | 58.3 | 51.3 |

It should be noted that NSP (100%), PSP (100%), NSP/PSP (50%/50%), and pure and blended pellet samples were used interchangeably throughout the article.

Since content analysis for this study focused on determining the percentage composition of major organic and elemental constituents of the pure and blended pellet samples, the data in Table 2 were interpreted in relation to variation in composition.

From Table 2, it can be noted that the major organic components of NSP (100%) differed from those of PSP (100%), indicating that woody biomass (like Norway spruce) is made up of different organic constituents in comparison to pea starch, whose organic constituents are composed of varying proportions of amylose and amylopectin. These two components (amylose and amylopectin) suggest that PSP (100%) is equally polysaccharide in nature. The balance of mechanical stability and durability offered by starch when blended with other materials in a pelleting process arises from its polysaccharide nature as a result of the chemical structures of its two major macromolecular components (amylose and amylopectin) [18]. In contrast to the pure pellet samples (100% each of NSP and PSP), the blend (NSP/PSP 50%/50%) showed a significant decrease in its organic constituents, a condition ascribed to component consolidation and the nature of the blended materials. This was basically viewed as a change in composition and structure that was construed to mean an alteration in chemical characteristics and an increase in strong intermolecular chain interactions. When starch is blended with materials such as wood, a product with modified properties and structure is obtained; the modification is facilitated by intermolecular chain interaction [43]. The alteration in the chemical characteristics of the blend (NSP/PSP 50%/50%) played a significant role in accelerating the tendency for particle-to-particle bonding because major organic components were consolidated upon blending. This was proven by complementary analytical techniques whose data are presented in the sections below. Nonetheless, the significant reduction in the major organic components of the blend also implied a reduction in the concentration of active bonding groups, a fact established by the FT-IR data presented in the next section.

For the elemental components of the pellet samples, the data in Table 2 indicate that C, H, and O constituted major elemental components of the pure and blended pellet samples of NSP and PSP used in this study. However, significant variations in the percentage composition of these elements could be noted, and the reason for the variation is the same as that given for the content of major organic components of the pellets.

It is noteworthy to mention that the percentage composition of the major organic and elemental components of the pure and blended pellet samples of NSP and PSP presented in Table 2 did not add up to 100% because the left-over fractions were ceded to minor components (such as extractives

and proteins) with percentage composition <3% and were considered insignificant; hence, they were not reported.

### 3.2. FT-IR Analysis

In addition to the goals of undertaking FT-IR analysis described in Section 2, data from the analysis also helped compare and establish differences in the mechanism of bonding between the pure and blended pellet samples relevant to quality. The FT-IR spectra of pure and blended pellet samples of NSP and PSP are presented in Figure 2. The data were obtained from the experimental analysis described in Section 2.4.2. To permit convenient comparisons, the spectra are presented as a single plot with the legends clearly displayed in the plot and spectral bands numbered for clarity.

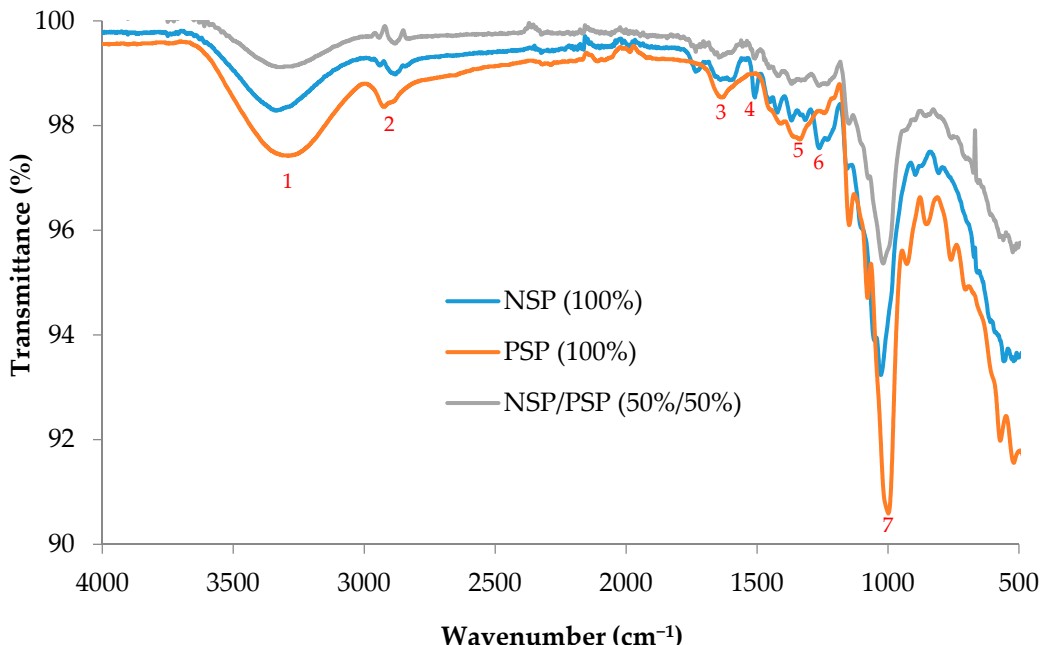

**Figure 2.** Fourier-transform infrared (FT-IR) spectra of pure and blended samples of Norway spruce pellet (NSP) and pea starch pellet (PSP).

Prior to interpretation of the FT-IR spectra in Figure 2, it is vital to mention that some spectral bands are more relevant than the others in the differentiation of the content (in terms of functional groups) of the pure and blended pellet samples. To avoid ambiguity, numbers were used to represent the most important bands considered relevant to the type of attraction forces holding particles of the pellets together, and to the quality of the pellets.

It is obvious from Figure 2 that the spectra were quite similar to each other, which does not necessarily mean the same characteristics. Using the pure pellet samples (100% each of NSP and PSP) as reference points, it is also apparent that the composite material (NSP/PSP 50%/50%) showed a synergistic effect. This synergistic behavior was more pronounced around 2893 to 1254 cm$^{-1}$ (bands 2–7), which were bands associated with active bonding groups such as C=O, C–O, C–O–C, and C–H, attributed to amylose and amylopectin, as well as cellulose, hemicellulose, and lignin in all three pellet samples [7]. All bands in this region (bands 2–7) had different intensities in the spectra of the three pellet samples. Some synergy can also be observed in band 2 of the spectra. The wide transmittance band around 3312 cm$^{-1}$ (band 1) for all samples was assigned to the hydroxyl group (–OH), an active bonding group responsible for band broadening due to a combination of intra and intermolecular hydrogen bonding. The presence and concentration of active bonding groups determine the type of attraction forces between individual particles of biomass pellets [7]. According to Popescu et al. [44], the broadening of the –OH band in IR spectra is often caused by the presence of

intra and intermolecular hydrogen bonds. However, the –OH group region of the IR spectrum is particularly useful for explaining patterns of hydrogen bonding, because each distinct –OH group offers a single stretching at a frequency that decreases with increasing strength of the hydrogen bonds, which are responsible for various properties of cellulose, hemicellulose, and lignin that are associated with NSP (100%) and NSP/PSP (50%/50%) [44].

Band intensities of samples in FT-IR analyses are true representations of the concentration of components [45]. Therefore, the active bonding groups previously listed absorbed light with greater intensity and indicated the presence of polysaccharides such as starch [46], leading to the more pronounced band intensity of PSP (100%). This suggests that PSP (100%) contained higher proportions of polysaccharides with greater amounts of active bonding groups (C–O and –OH groups) capable of stronger dipole–dipole attraction and hydrogen bonding (which are two types of intermolecular forces of attraction) as compared to pure NSP (100%) and NSP/PSP (50%/50%). Nonetheless, in the spectrum of NSP (100%), the transmittance band around 1275 and 1478 cm$^{-1}$ (bands 4 and 6) was due to distance-dependent interactions associated with van der Waals contact distances as a result of groups such as C=O and C–H, which are known to form bands within the wavelengths specified above. In addition to being active bonding groups, these groups are equally polar in nature because of their ability to form different types of attraction forces [7]. Thus, for the blend (NSP/PSP 50%/50%), it was presumed that its particles were held together by strong hydrogen bonding because of the presence of the C=O group, which is a polar functional group capable of initiating structural changes by introducing more intermolecular forces of attraction.

With respect to the quality of the pellets, defined in this study in terms of strength and burning efficiency, it is fair to allude that, in relation to the strength of the pellets, PSP (100%) seemed to be a higher-quality pellet than NSP (100%) and NSP/PSP (50%/50%). This was because of its more pronounced band intensity, which was construed to mean greater proportions of polar functional groups that were capable of forming a combination of dipole–dipole attraction and hydrogen bonding between particles. A union of these two attraction forces creates stronger bond energies/bond strength, whereas the functional groups associated with NSP (100%) formed more of van der Waals attraction forces than hydrogen bonding between particles of the pellet, which may increase the possibilities of the formation of fines. Particles of the blend (NSP/PSP 50%/50%) were more connected by a combination of multiple forces of adhesion and cohesion, some of which may have reduced bond strengths that were easily broken. These findings are in agreement with the fact that the quality of biomass pellets, among other factors, depends on the type and strength of attraction forces between particles, as alluded by Kaliyan [41]. In view of this, the order of strength of the pellets was as follows: PSP (100%) > NSP/PSP (50%/50%) > NSP (100%). For the quality of the pellets in terms of burning efficiency, the thermal analysis data presented in Sections 3.3 and 3.4 has provide the needed information.

It is worth mentioning that spectral interpretation from FT-IR analysis of samples is non-exhaustive. Thus, for clearer spectral interpretation and understanding, the peak absorption range of different functional groups can be found in Reference [47].

### 3.3. TGA Analysis

The determination of the modification range temperature, the effects of blending, and the comparative assessment of the mechanism of bonding relevant to pellet quality are among important instances where thermal analysis addresses specific needs. One of the tools proven to address these needs is a thermogravimetric analyzer (TGA). Thus, knowledge of the major compositional changes that take place as biomass undergoes thermal treatment can facilitate understanding of its behavior in pretreatment processes including in pelleting processes [48]. Figure 3 shows the TGA thermograms of the pure and blended pellet samples with the focus area clearly marked for less ambiguous interpretation.

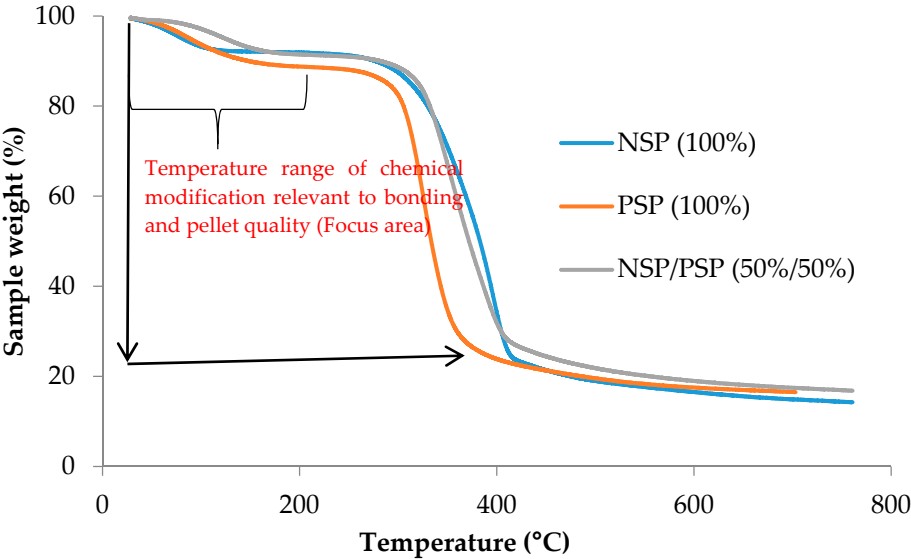

**Figure 3.** Thermogravimetric analysis (TGA) thermograms of pure and blended samples of NSP and PSP.

Interpretation of the TGA plot in Figure 3 was limited to the focus area marked in the plot. Events beyond the focus area were not considered because, according to some studies [3,5,7,8,41,49,50], the optimum temperature range for pelleting biomass falls between 50 and 150 °C. Within this temperature range, flow and combustion characteristics relevant to bonding and pellet quality are usually formed [5,7,8]. However, it is important to bear in mind that thermal degradation of a sample under TGA proceeds differently in terms of the pattern, based on how the reduction in sample weight occurs as the sample undergoes combustion [51]. This means that the weight change of the sample (increase or decrease in weight) as temperature increases depends on a host of factors, including the type of sample, its stability, and analysis conditions [33,35,52]. Having said this, while the thermal behavior of NSP (100%) was controlled by its major organic components (cellulose, hemicellulose, and lignin), that of PSP (100%) was equally influenced by its primary organic macromolecular constituents (amylose and amylopectin), and the combustion behavior of the blend (NSP/PSP 50%/50%) was influenced by a combination of the contents of both materials (Norway spruce and pea starch).

Eye-tracking TGA curve information from Figure 3 shows that the weight change of all three pellet samples was completed at about 760 °C. The first noticeable feature, however, was the reduction in weight of the samples as temperature increased (indicated by the arrows), a condition which forms the core of the thermal performances of biomass materials [7,33]. Another observable feature is the differences in the degradation patterns of the curve, particularly around the focus area (focus point). This point represents the temperature range of chemical modification in which water molecules and other small molecular species relevant to bonding are liberated [7]. The release of moisture began at almost the same temperature for all samples (around 45 °C), but the temperature at which molecular species began to flow was slightly different between the pure and blended pellet samples, an indication of differences in thermal behavior attributed to compositional disparity. This temperature for NSP (100%) was about 55 °C, around 65 °C for PSP (100%), and ca. 60 °C for NSP/PSP (50%/50%). Shortly after moisture evaporation and the release of molecular species, critical evaluation of the plot showed that the degradation temperature for NSP (100%) relevant to pelleting began at ca. 85 °C, while that for the blend (NSP/PSP 50%/50%) started at about 75 °C, with that for PSP (100%) beginning at a moderately higher temperature of above 90 °C. This typically means that the release of the bonding components of PSP (100%) required more energy than those of NSP (100%) and NSP/PSP (50%/50%), a condition believed to be attributed to higher concentrations of polar functional groups, particularly the C–H group (from band 7 in the FT-IR spectra in Figure 2) in the structure of PSP (100%) with

greater bond strength and bond energy as compared to NSP (100%) and NSP/PSP (50%/50%). The bond strength and bond energies of different functional groups, as well as those of various attraction forces, can be found in References [53,54]. Atoms of polar functional groups are held together by a combination of strong dipole–dipole attraction and intermolecular forces that require more energy to be broken, resulting in the thermal behavior of PSP (100%) being different than that of NSP (100%) and NSP/PSP (50%/50%), for which the energy required for bond breaking would be much less. The more concentrated these groups are in a material, the greater is the energy required to break the bonds holding atoms of these groups together, which can also be construed to mean that the stronger a bond is, the higher the energy required is to break the bond [7,50,54]. A review of the literature [53,54] also showed that the bond energies of different functional groups and those of different types of attraction forces vary.

For the quality of the pellets in terms of burning rate and efficiency of combustion, this analysis (TGA) established that NSP (100%) may be a better quality pellet in comparison to PSP (100%) and NSP/PSP (50%/50%) because of its lower modification temperature, which was construed to mean faster burning velocity that translates into better combustion efficiency due to greater contact between the pellet sample (NSP 100%) and the heating environment. Biomass combustion efficiency is a function of good contact between the heating environment and the biomass [55]. According to Gil et al. [56], the burning rate of pure biomass pellets, which is determined by temperature, is usually higher than that of their blends. Furthermore, judging by the FT-IR data of the pellets presented in Figure 2, there were higher percentages of oxygen-containing polar functional groups for PSP (100%). This puts PSP (100%) at a great disadvantage as a feedstock in thermal energy conversion systems due to poor combustion-related issues that may significantly increase the operating and maintenance costs of the energy systems. This is because oxygen-containing polar functional groups like the –OH groups are hydrophilic in nature and have the ability to retain moisture, thereby increasing the viscosity of starch micelles (starch gelatinization) [57,58]. The issues of starch gelatinization are explained in greater detail in a correlative thermal analysis data presented in the section below. In the case of the blend (NSP/PSP 50%/50%), however, the oxygen-containing polar functional groups may have been compromised or compensated for by the cellulose, hemicellulose, and lignin contents of Norway spruce. In view of these results, therefore, the order of quality of the pure and blended samples of NSP and PSP in terms of burning rate and efficiency of combustion was as follows: NSP (100%) > NSP/PSP (50%/50%) > PSP (100%).

It is vital to mention that there are no standard methods to determine combustion efficiency; hence, data from thermal analysis of biomass pellets aimed at burning efficiency cannot be compared without uncertainties [59].

*3.4. DSC Analysis*

According to Back [39], sufficient bonding areas are created when wood components are plasticized above their glass transition temperature ($T_g$). In addition, since combustion is the final destination of most biomass pellets, an ultimate method for testing the quality of biomass pellets is thermal analysis, because good-quality pellets are expected to burn easily and generate less complex degradation patterns [60,61]. Figure 4 presents the DSC plot of pure and blended samples of NSP and PSP.

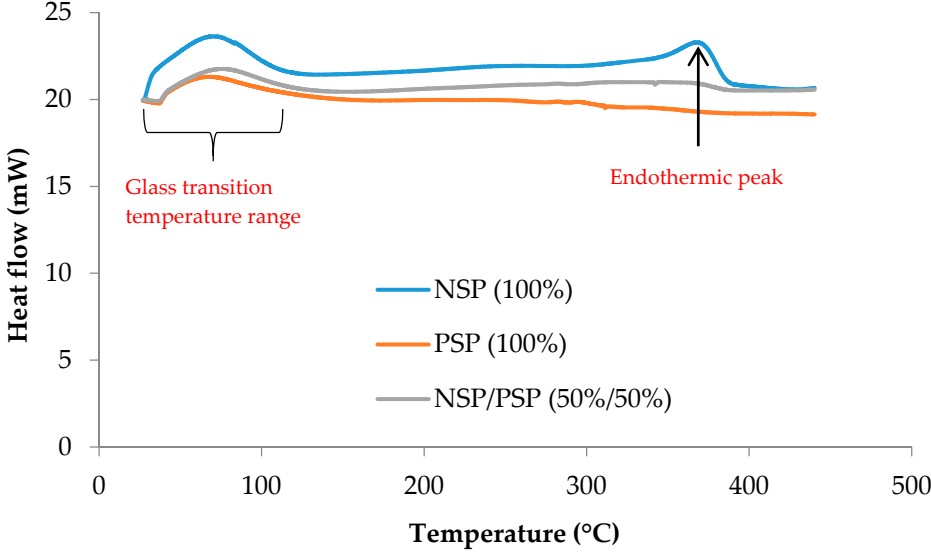

**Figure 4.** Differential scanning calorimetry (DSC) thermograms of pure and blended samples of NSP and PSP.

A step change in the heat flow of the DSC thermogram presented in Figure 4 constituted the first observable feature of the plot and represented the glass transition ($T_g$) range temperature for all samples. $T_g$ conditions facilitate the deformation of particles, reduce viscosity, and increase the movement of natural binding components [41]. According to the DSC plot in Figure 4, these temperature conditions fell between 45 and 85 °C for NSP (100%), between 45 and 80 °C for PSP (100%), and between 45 and 75 °C for NSP/PSP (50%/50%). Moisture is one of the most useful agents that can act as a lubricant during pelleting of biomass materials because it strengthens and promotes bonding through a combination of attraction forces by increasing the particle contact area [3,41,62]. Just like the previous thermal analysis data, transition at 45 °C in the DSC plot typically indicated the moisture evaporation from all samples. The $T_g$ range of the samples under DSC analysis were consistent with the previous modification temperature range presented in Section 3.3, except that, in this analysis (DSC analysis), the maximum $T_g$ for PSP (100%) occurred at a much lower temperature (80 °C) than its previous maximum modification temperature of around 90 °C in the previous section. The complexity of the thermal behavior of starch and differing analysis conditions may be responsible for the difference in the maximum modification and transition temperatures between this thermal analysis and the previous for 100% PSP. The behavior of starch under thermal conditions is much more complex than other thermoplastic materials like wood because the physicochemical changes that occur during combustion of starch or products containing starch involve the motion of water, gelatinization, glass transition, and melting, as well as the change of crystal structure and molecular degradation, which are thermal events that are dependent on material moisture content, and the water contained in starch is often not stable during combustion [63].

The DSC plot in Figure 4 also indicated that 100% NSP displayed two endothermic peaks, with its first in the $T_g$ range, and its second way beyond the $T_g$ range (between 350 and 370 °C). However, further interpretation of the DSC plot beyond the Tg range for all samples was less of a focus because the study aimed to establish differences in the mechanism of bonding between the pellet samples relevant to quality, and the $T_g$ is considered the temperature range at which natural binders relevant to bonding are activated [6,41]. Therefore, inter-diffusion of the natural binders in 100% NSP increased between adjacent particles to facilitate bonding and the formation of solid bridges at its $T_g$. The polymeric constituents of the blend (NSP/PSP 50%/50%) were released faster than those of the pure pellet samples (NSP 100% and PSP 100%) because of lower $T_g$ and the presence of water vapor created by moisture. Differences in $T_g$ between the pure and blended pellet samples can be traced back to

their content of polar functional groups (Figure 2) including the oxygen-containing groups whose concentrations were higher with stronger bond energies in PSP (100%) than NSP (100%) and the blend (NSP/PSP 50%/50%).

For the quality of the pellets in terms of combustion efficiency, which was considered in this analysis (DSC analysis) from the viewpoint of burning and heat flow rates, NSP (100%) seemed to dominate because of the two endothermic peaks formed in its DSC plot (Figure 4). The formation of the two endothermic peaks and the modification temperature for NSP (100%) from the previous thermal analysis data (55 °C as compared to 60 °C and 65 °C for NSP/PSP 50%/50% and PSP 100%, respectively) were evidence of increased burning and heat flow rates that were facilitated by auto-oxidation reactions because of the presence of oxygen-containing functional groups in the structure of NSP (100%). This means that the burning and heat production rates of NSP (100%) were significantly higher in comparison to those of PSP (100%) and NSP/PSP (50%/50%). Although, PSP (100%) may have greater concentrations of oxygen-containing polar functional groups, there is the possibility of the formation of hydrogen bridges at elevated temperatures because of increased probabilities of the promotion of retrogradation reactions that may result in the formation of thickened paste, which reduces combustion efficiency. However, this thickened paste can facilitate particle-to-particle bonding to form pellets with increased strength due to the presence of rigid irreversible gel [64]. The concentrations of the oxygen-containing polar functional groups in the blend were compromised by mixing the two materials at a 50/50 ratio. However, because of the balance between the ratio of the two materials in the blend (NSP/PSP 50%/50%), and perhaps the higher percentage of carbon in NSP (100%) in comparison to PSP (100%), combustion issues would be minimized when the blended pellet is used as feedstock in thermal conversion systems. Therefore, the order of quality of the pellets in terms of combustion efficiency was as follows: NSP (100%) > NSP/PSP (50%/50%) > PSP (100%).

### 3.5. SEM Analysis

When biomass is heated beyond 40 to 50 °C, its morphology changes, creating flow characteristics that are not just determined by temperature but also by the structure of the biomass, such as surface regression and major fragmentations that are relevant to bonding because natural binders contained in the biomass are activated upon increasing temperature [6,7,34,65]. Figure 5 presents cross-sections of the SEM images of pure and blended samples of NSP and PSP obtained under the same measurement conditions such as those inscribed in the images.

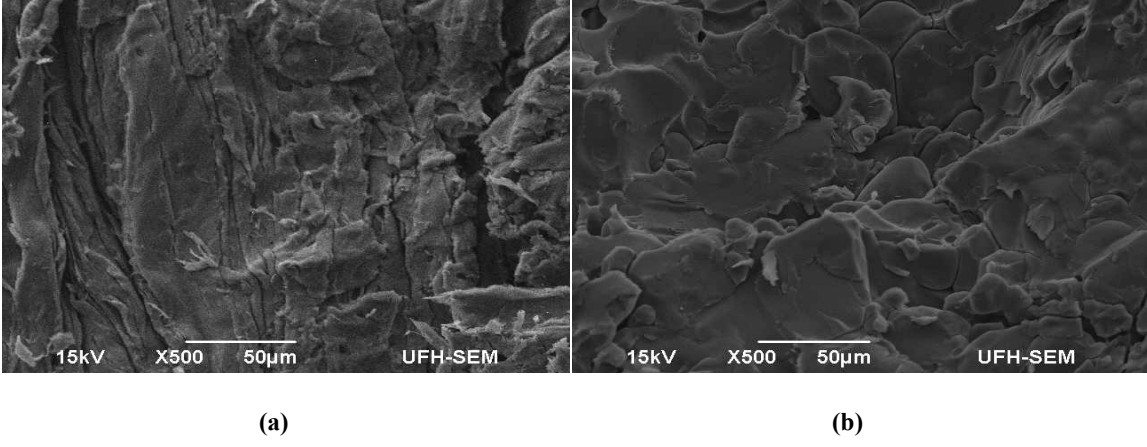

|　　　(a)　　　|　　　(b)　　　|

**Figure 5.** *Cont.*

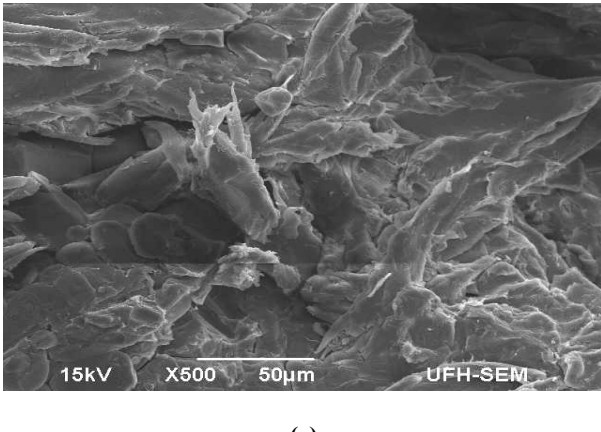

**(c)**

**Figure 5.** The SEM images of pure and blended samples of NSP and PSP: (**a**) NSP (100%); (**b**) PSP (100%); (**c**) NSP/PSP (50%/50%).

Figure 5a–c typically show the fracture surfaces of the pure and blended samples of NSP and PSP. The nature of the fracture surfaces was an indication of the production of pellets with differences in visual appearance. It could also be noted that the surfaces of the pellets were quite rough with interconnected particles and voids that characterized the image of PSP (100%), which were believed to have been created by pelleting process conditions such as compression force and temperature. The observable particle-to-particle interconnection was equally an indication of the promotion of adhesion by increased molecular contact between particles, which also represented the presence of short-range attraction forces such as intra and intermolecular forces including van der Waals forces, attributed to the presence of increased polar functional groups, free chemical bonds, and hydrogen bridges [41,66,67]. It could also be noted from the images that particles were clustered and densely packed, an indication that the materials were milled to relatively small particle sizes of <1 mm prior to pelleting. Surface morphology of the pellet samples equally showed conspicuous differences in the pattern of roughness, particularly between the images of NSP (100%) and PSP (100%), whose SEM image, as earlier stated, showed more void spaces that were believed to have been caused by a pelleting spring-back effect due to the highly reactive oxygen-containing polar functional groups associated with PSP (100%). This spring-back effect was an indication of interlocking of fibers and that bonding in 100% PSP was most likely due to dipole–dipole and hydrogen bonding. According to Mani et al. [68], the compaction of biomass grinds occurs in different stages, such as the formation of void spaces that facilitates inter-particle contacts. Surface roughness enhances bond strength, which is intimately linked to bond energy [69–71]. Therefore, the pattern of roughness is a strong indication of differences in the mechanism of bonding between the particles of the pure and blended pellet samples. However, the voids observed in the image of PSP (100%) were no longer visible in the SEM image of NSP/PSP (50%/50%), a clear evidence of component consolidation and an indication that components of the two materials (Norway spruce and pea starch) interacted to create stronger unions, perceived as a type of intermolecular bonding connected to covalent and hydrogen bonding [72]. Furthermore, the particles of PSP (100%) were more spherical in shape than those of NSP (100%), and also appeared complemented in the blend (NSP/PSP 50%/50%) (Figure 5c). Spherical particles have higher surface areas that are more densely packed; as a result, they are likely to form pellets with greater strength [49].

With regard to the quality of the pellets, the SEM images did not provide clear morphological features that could be linked to differences in the quality of the pellets, whether in terms of strength, bulk density, or of any other quality parameter for that matter; hence, it was difficult to draw conclusions on the quality of the pellets from this analysis.

## 4. Discussion

In this study, pure and blended fuel pellets made from Norway spruce and pea starch were analyzed using advanced analytical instruments able to provide information beyond what is visible to the naked eye in order to establish differences in particle bonding mechanism relevant to quality. The results obtained were comprehensively interpreted from a structural chemistry perspective and used to answer questions relating to how particles of pure and blended pellets made from two different biomass materials combine to form pellets, and the source of inter-particle bonding. As previously stated, assessing biomass for any conversion or pretreatment process requires an understanding of its basic composition [1,3,32]. Therefore, compositional analysis data determined that pure and blended samples of NSP and PSP assessed in this study contained various proportions of major organic and elemental constituents, which were vital to understanding the nature of the materials that were pelletized, and which were equally significant in comprehending how the major components combine to form pellets.

From the FT-IR data, it was established that the pure and blended samples of NSP and PSP contained varying concentrations of polar and non-polar functional groups that played a significant role in providing information related to differences in the mechanism of bonding and the type of attraction forces between individual particles relevant to quality in terms of strength only. It was determined that bonding in NSP (100%) was mostly due to van der Waals attraction forces, while, for PSP (100%), bonding was attributed to a combination of dipole–dipole and hydrogen bonding. In NSP/PSP (50%/50%), particles were held together by a combination of forces that included hydrogen bonding, dipole–dipole interaction, and slight van der Waals attraction forces. From previous studies [7,41,73,74], it was determined that the quality of biomass pellets, among other factors, is dependent upon the type and strength of attraction forces between individual particles; hence, FT-IR analysis aided the determination of the quality of the pellets in terms of strength based on the theory of functional groups and the strength of the forces acting between individual particles, for which the order of strength of the pellets was given in the last paragraph of Section 3.2. In this study, therefore, spectral deconvolution from FT-IR analysis provided significant information not just about the type of attraction forces between individual particles of the pellets, but also bond strength, which helped to establish the quality of the pellets without having to rely on special quality assessment tools.

Thermal analysis data from TGA and DSC of the pure and blended samples of NSP and PSP suggested that the behavior of the materials (Norway spruce and pea starch) under pelleting temperature was controlled by major components of the pellets, and that temperature differences between the pellets played a key role in determining when molecular species relevant to bonding were released. This means that how particles of the materials combined to form pellets during pelleting was determined by temperature, as noted by the degradation patterns of the resulting thermograms from TGA and DSC analyses. In relation to the quality of the pellets, which was defined in terms of burning rate, heat flow rate, and combustion efficiency, NSP (100%) dominated because of reasons given earlier in Sections 3.3 and 3.4. Differences in the modification temperatures between the pure and blended pellet samples also suggested that pelleting pure pea starch would require more energy than pelleting pure Norway spruce, and a 50/50 blend of these two materials would most likely show a balance in energy consumption during pelleting. Nevertheless, a comparative energy consumption analysis of pelleting pure and blended Norway spruce and pea starch must be conducted to establish the energy needed. Judging by the strength of the forces holding its particles together, PSP (100%) was equally a good-quality pellet in terms of strength; however, its poor thermal conductivity as a result of its high oxygen to carbon (O–C) ratio with the ability to retain moisture renders the pellet unsuitable as feedstock in thermal conversion systems for energy production purposes. Biomass feedstock with a high O–C ratio reduces the heating value of the biomass and causes an increased amount of smoke with greater formation of water vapor, as well as significant energy losses when such biomass is used as feedstock in thermal energy production systems [35,52,75–78].

From SEM analysis, it was established that the pattern of surface roughness of the pellets and the mode of interconnectivity of particles, observed from the SEM images of the pellets, offered a strong indication of differences in bonding mechanisms between the pure and blended pellet samples, which basically showed that major components from two or more different materials behave differently under certain pelleting conditions such as temperature and compression force, and that their behavior is affected by the content and concentrations of polar functional groups. According to a previous study [7], functional groups confer specific properties and act differently under certain pelleting conditions such as those presented in Table 1. With respect to the quality of the pellets, however, morphological features from SEM analysis could not establish any observable structural changes of importance to quality between the pellets, no matter how quality was defined.

Much still remains to be understood about differences in the mechanism of bonding between pure and blended pellets made from different biomass materials and the chemistry of not just the strength of the pellets, but also of their combustion. Although there are many similarities in the composition of biomass, there are equally huge differences; hence, understanding how components of different biomass combine to form good-quality pellets is made difficult by the variability in the composition of biomass and the multiple events that occur as biomass undergoes pelleting. Given this difficulty, however, it is vital to generate a full understanding of the structural and chemical properties of pelletized biomass relevant to good quality in all ramifications. As such, further studies are required on the bonding characteristics of pure and blended biomass before pelleting, employing contemporary analytical techniques so that a comparison can be made with this study in order to achieve better understanding of how milled biomass is transformed to pellet, as well as the source of inter-particle bonding in pure and blended biomass pellets made from different biomass resources. The ratio of the blended materials should also be widely varied and pelletized in order to conclusively establish differences in the mechanism of bonding relevant to quality between the varied biomass pellets. To corroborate evidence of quality of the pure and blended pellets provided by the state-of-the-art analytical instruments used in this study, it is recommended that standard quality assessment tools be used.

In light of what was investigated in this study, it is fair to allude that the significance of the study is related to the information it adds to differences in the mechanism of bonding between pure and blended biomass pellets from two different materials and how this correlates to the quality of pellets.

## 5. Conclusions

Based on the findings of the study, the following conclusions were drawn:

- Satisfactory information was obtained from compositional analysis of pure and blended NSP and PSP, which established that the pellets contained varying proportions of organic and elemental components that were particularly responsible for their pelleting behavior under the conditions presented in Table 1. Data from subsequent analytical instruments also established that the major components of the pellets exhibited varying bonding attributes that were used to predict their quality in terms of strength and combustion efficiency.
- The data from FT-IR analysis showed that differences in the concentration of polar functional groups between the pure and blended pellet samples explained the type and strength of attraction forces between individual particles of the pellets, with PSP (100%) exhibiting better quality in terms of strength than NSP (100%) and NSP/PSP (50%/50%) due to the strength of the forces acting between its particles. Therefore, the presence of functional groups and the variation in intermolecular forces between particles of the pure and blended pellets were the sources of differences in their particle bonding mechanism.
- Temperature was a determinant factor of how particles of the pure and blended pellet samples combined to form pellets (differences in bonding mechanism between particles of the pure and blended pellets). This was because of differences in the modification/transition temperature where molecular species relevant to bonding were considered to be released. For instance, the release of

the bonding components of PSP (100%) began at a much higher temperature compared to NSP (100%) and the NSP/PSP (50%/50%) blend, a condition which suggested that flow characteristics of polymeric constituents relevant to bonding were delayed for PSP (100%), thus slowing its mechanism of particle bonding.

- Thermal analysis results from TGA and DSC also established that NSP (100%) had better burning efficiency in comparison to PSP (100%) and the NSP/PSP (50%/50%) blend.

- The variation in surface roughness and interconnectivity of particles that was noted from the SEM images of the pellets was a strong indication of differences in their particle bonding mechanism, and evidence of the application of compression force. These conditions were facilitated by the presence of functional groups, which were assumed to act differently under certain pelleting process conditions such as those presented in Table 1.

- Due to the formation of void spaces between its particles and the lack of inter-particle polymer bridges, bonding in PSP (100%) may be due to dipole–dipole forces and interlocking of fibers in comparison to van der Waals forces in NSP (100%) and hydrogen bonding in the NSP/PSP (50%/50%) blend.

- The three pellets studied all showed evidence of good quality in terms of strength; however, in terms of combustion efficiency, only two of the pellets (NSP 100% and the NSP/PSP 50%/50% blend) were considered good-quality pellets for reasons given earlier.

**Author Contributions:** Conceptualization, A.I.A., J.B., and S.F.; methodology, A.I.A., J.B., and S.F.; investigation, A.I.A.; data curation, A.I.A.; writing—original draft preparation, A.I.A.; writing and editing, A.I.A., J.B., and S.F.; resources, E.B.F. and P.N.

**Funding:** This research was funded by the Swedish Agency for Growth through the FOSBE project with grant number 20201239, and the APC was funded by Karlstad University Library.

**Acknowledgments:** The authors wish to thank the Division of Environmental and Energy Systems within the Department of Engineering and Chemical Sciences of Karlstad University for their administrative support.

**Conflicts of Interest:** The authors declare no conflicts of interest.

## Abbreviation

| Abbreviation | Definition |
| --- | --- |
| NSP | Norway spruce pellet |
| PSP | Pea starch pellet |
| TGA | Thermogravimetric analysis |
| DSC | Differential scanning calorimetry |
| EDX | Energy-dispersive X-ray spectroscopy |
| SEM | Scanning electron microscopy |
| (FT-IR) | Fourier-transform infrared spectroscopy |
| IR | Infrared |
| $T_g$ | Glass transition |

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
