# Peer review of "Characterization of Pure and Blended Pellets Made from Norway Spruce and Pea Starch: A Comparative Study of Bonding Mechanism Relevant to Quality"

_energies, doi:10.3390/en12234415_

Round 1

Reviewer 1 Report

The paper has been beautifully well written with no errors detected starting from the abstract all the way to the conclusion. Based on the above, it should be accepted the way it has been presented.

Author Response

We appreciate the reviewer’s kind remarks and recommendation for our manuscript to be accepted for publication.  

Reviewer 2 Report

The investigations implemented are interesting but not well reported. Major revision is needed. It appears that many statements are contradictory to the content of the Figures reported. Also the structure of the presentation needs to be modified.

More specific comments:

The first 40% of the Abstract contains introductory statements that do not belong to an Abstract. Abstract could mention methods, but should focus to results and their eventual implications.

It appears that lines 118 to 120 give statements regarding accomplishments of the paper, written in the past tense. Such statements hardly can be located under the title “Purpose of the study”.

Line 186, along with Table 1, provides a statement of optimal pelleting conditions, without any reference to any optimization process.

Lines 208 to 211 appear to contain a combination of introduction and discussion. However, the text is under the heading “Materials and methods”. The second sentence appears meaningless – something missing?

Apparently lines 216 to 221, discussing TGA results reported in Fig. 3, are not in their correct position within the manuscript. Is Fig. 3 the second Figure mentioned in the paper?

Lines 230 to 232: an emitted X-ray was bombarded and then collected?

Lines 243 to 245: this paragraph should contain presentation of methods, not value discussion.

Lines 322 to 326. This paragraph is not understood by the reviewer.

Figure 2 shows very interesting behavior. The composite material shows a synergistic effect: absorbance is consistently greater than in any of the pure components. This observation is serious disagreement with the related text by the Authors. Maybe the axis labelling is incorrect? Should the vertical axis label be “Transmittance”, or something like that?

Lines 396 to 444 appear to contain discussion.

Figure 3 appears astonishing. Almost all of the weight of specimens is lost at room temperature. However, much of it is regained when temperature is increased. The text below the Figure agrees with the Figure. However, collecting some courage, I dare to assume that reality is something else: possibly the axis label has to be changed, or the axis reversed, and the related text corrected.

Lines 494 to 513 appear to contain discussion and conclusions, instead of presentation of results. The reviewer is unable to find any evidence that would justify the statements.

Figures 2 and 3 appear to be rather essential contents of the paper. The contents of Fig. 2 being contradictory to the related text, and the contents of Fig. 3, along with the related text, being suspected by this reviewer to be incorrect reporting of measurements, this reviewer does not think he would have much of a possibility of understanding further discusson in the paper. For now, the review terminates at line 513.

Author Response

Comments and Suggestions for Authors

The investigations implemented are interesting but not well reported. Major revision is needed. It appears that many statements are contradictory to the content of the Figures reported. Also the structure of the presentation needs to be modified.

Response: We appreciate the constructive criticism of the reviewer with regards to the reporting of what was investigated in our work. The manuscript has been thoroughly revised with content of the reported Figures made as explicit as possible and contradictions avoided. The structure of the presentation has also been modified for better readability and understanding. 

More specific comments:

Point 1: The first 40% of the Abstract contains introductory statements that do not belong to an Abstract. Abstract could mention methods, but should focus to results and their eventual implications.

Response 1: The abstract has been re-written and re-worded in a way that now reflects the main results of the work.  

Point 2: It appears that lines 118 to 120 give statements regarding accomplishments of the paper, written in the past tense. Such statements hardly can be located under the title “Purpose of the study”.

Response 2: The statements in lines 118 to 120 have been deleted.

Point 3: Line 186, along with Table 1, provides a statement of optimal pelleting conditions, without any reference to any optimization process.

Response 3: A reference to an optimization process have been provided in line 186 (now line 208 to 210), and in Table 1.

Point 4: Lines 208 to 211 appear to contain a combination of introduction and discussion. However, the text is under the heading “Materials and methods”. The second sentence appears meaningless – something missing?

Response 4: From our point of view however, the text in question is rather under the subheading ‘’2.4.1. Content analysis’’ (lines 231 to 235 of the revised manuscript) and reflects introduction of the section before presentation of what was undertaken in the section. The text was written to justify the need to conduct content analysis, which to us was perceived as a standard practice in writing articles of this nature. However, the sentences including those that appear meaningless, have been rephrased for better readability and understanding. We’d like to reiterate that pelleting is a type of pre-treatment method undertaken on biomass for the purpose of improving the properties of the biomass. It alters the functionality of the constituents of the biomass. Therefore, since the study aims to characterize pure and blended pellet samples, it was necessary to determine major components and establish the role each played in the mechanism of bonding and quality of the pellets. As such, it was essential to introduce the section using terms that would justify the need for what was undertaken.

Point 5: Apparently lines 216 to 221, discussing TGA results reported in Fig. 3, are not in their correct position within the manuscript. Is Fig. 3 the second Figure mentioned in the paper?

Response 5: The discussions in lines 216 to 221 (deleted lines 241 to 246 of the revised manuscript) have been removed and placed in its right position within the manuscript, specifically lines 291 to 299. Furthermore, the contradictions about the Figures mentioned in the paper have been made clear by the sentences placed in their right positions (lines 291 to 299).  

Point 6: Lines 230 to 232: an emitted X-ray was bombarded and then collected?

Response 6: The sentences in lines 230 to 232 (now lines 256 to 259) were made as clear as possible. More information was briefly provided to support what was described. The EDX is an instrument with an analytical capability that can be coupled with several other instruments such as SEM, TEM, etc. It is used to quantify the weight fractions of the elemental components of a sample and its principle of operation is such that the instrument collects X-rays that are emitted from a sample after the sample has been bombarded by an electron beam. The collected X-rays are characteristic of the atomic structure of the elements contained in the sample. Generally, the energy of the X-rays emitted depends on the sample under analysis. The following non-exhaustive list of articles/links could help the reviewer better understand the mode of operation of the EDX instrument:

Standard operating procedure for sample preparation and analysis of PMIO and PM2.5 samples by Scanning Electron Microscopy. A manual by the Research Triangle Institute (RTI), 2008 https://www.sciencedirect.com/topics/materials-science/energy-dispersive-x-ray-spectroscopy https://en.wikipedia.org/wiki/Energy-dispersive_X-ray_spectroscopy

Point 7: Lines 243 to 245: this paragraph should contain presentation of methods, not value discussion.

Response 7: We respectfully disagree with the reviewer about the statements in lines 243 to 245 (now lines 271 to 273). This is because the paragraph justifies the need for what was undertaken (FT-IR analysis), which was perceived as a key reason why FT-IR analysis was conducted on the pellets. From our point of view however, investigations involving complex topics such as the mechanism of bonding in biomass pellets, which is still a much debated topic in biomass pellets research, standard practice requires that information be provided as much as possible with clear justifications of the selected methods of research and their associated results. The paragraph in question has been rephrased for better readability.

Point 8: Lines 322 to 326. This paragraph is not understood by the reviewer.

Response 8: The paragraph in lines 322 to 326 (now lines 362 to 364) has been rephrased for better understanding.

Point 9: Figure 2 shows very interesting behavior. The composite material shows a synergistic effect: absorbance is consistently greater than in any of the pure components. This observation is serious disagreement with the related text by the Authors. Maybe the axis labelling is incorrect? Should the vertical axis label be “Transmittance”, or something like that?

Response 9: We agree with the reviewer about the synergistic effect in the behaviour of the materials. However, the aim of the FT-IR analysis was to identify the type of attraction forces acting between individual particles of the pellets (which were achieved through identification of key functional groups and the concentration of these groups) and the relevance of these forces to the mechanism of bonding and quality of the pellets. Therefore, interpretation of the FT-IR spectra of the pellets was in relation to differences in the type and concentration of functional groups, type and strength of attraction forces, as well as the influence of these in bonding and quality of the pellets. The latter factor (pellet quality) was determined based on the previous factors (type and concentration of functional groups, type and strength of attraction forces). Spectral deconvolution gave significant information about bond strength, which helped to establish the quality of the pellets without having to rely on special quality assessment tools. All these were made clear in our interpretation of the FT-IR spectra of the pellets presented in Figure 2. Nonetheless, ‘’absorbance’’ mode measurements in FT-IR analysis have a significant advantage in the sense that it allows the measurement of more intense bands in comparison to ‘’transmittance’’ measurements, which display distorted and/or cut off maxima. Nevertheless, FT-IR analysis is dependent on personal preference.  

Point 10: Lines 396 to 444 appear to contain discussion.

Response 10: Line 396 to 444 (now lines 437 to 488) is a continuation of the spectra interpretation and its implication to the quality of the pellets. This discussion is a part of the in-depth analysis of the results supported by detailed literature information. However, some texts have been moved from their previous positions to where they are meant to be. For example the text formerly in lines 469 to 471 of the revised manuscript is now in lines 486 to 488.

Point 11: Figure 3 appears astonishing. Almost all of the weight of specimens is lost at room temperature. However, much of it is regained when temperature is increased. The text below the Figure agrees with the Figure. However, collecting some courage, I dare to assume that reality is something else: possibly the axis label has to be changed, or the axis reversed, and the related text corrected.

Response 11: The TGA plot in Figure 3 highlighted a focus point that was discussed in relation to compositional changes caused by temperature and the relevance of these changes to the mechanism of bonding and quality of the analysed pellets. Using PSP (100%) as reference, the arrows in the plot were added to indicate that as temperature increases from room temperature, the samples lose weight as a result of the release of excess amount of volatiles hence the rapid weight loss between 80°C and 25°C (the room temperature). This section of the plot was not discussed in relation to mechanism of bonding and the quality of the pellets. However, since the study aims to understand the mechanism of bonding in pure and blended pellets made from two different materials, it was necessary to consider the most suitable temperature for pelleting of biomass, and according to literature information, optimum pelleting temperature is in the range 50 to 150°C, which happens to be the temperature range captured in our discussion of the TGA results of the pellets.

We refer the reviewer to lines 500 to 503 of the revised manuscript.

It is vital to mention that TGA remains an analytical technique used to determine the thermal behaviour of a material and the fraction of volatile constituents of the material by monitoring the weight change that occurs as the material is heated at a constant rate. A plot of weight change vs temperature is usually obtained and the plot has a characteristic shape for biomass, whether or not the biomass is pelletized. Nonetheless, presenting information that is already known about analytical instruments and their associated data will render the manuscript ambiguous. Just like FT-IR analysis, interpretation of the data from TGA is equally dependent on personal preference. To help the reviewer better understand TGA data/plots of biomass materials/pellets, a non-exhaustive list of articles are provided below:

Anukam A.I., Berghel J., Famewo E.B., Frodeson S. Improving the understanding of the bonding mechanism of primary components of biomass pellets through the use of advanced analytical instruments. Journal of Wood chemistry and Technology 2019, 1 – 18. Anukam A., Mamphweli S., Reddy P., Okoh O., Meyer E. An investigation into the impact of reaction temperature on various parameters during torrefaction of sugarcane bagasse relevant to gasification. Journal of Chemistry 2015, 1 – 12. de Jonga W., Pironea A., Wo´jtowicz M.A. Pyrolysis of miscanthus giganteus and wood pellets: TG-FTIR analysis and reaction kinetics. Fuel 2003, 82, 1139 – 1147. Garcia-Maraver A., Terron L.C., Zamorano M., Ramos-Ridao A.F. Thermal events during the combustion of agricultural and forestry lopping residues. Materials and processes for energy: communicating current research and technological developments 2013, 407 – 413. Haobin P., Li Y., Li Y., Yuan F., Chen G. Experimental investigation of combustion kinetics of wood powder and pellet. Journal of Combustion 2018, 1 – 7.

Point 12: Lines 494 to 513 appear to contain discussion and conclusions, instead of presentation of results. The reviewer is unable to find any evidence that would justify the statements.

Response 12: Lines 494 to 513 (now lines 538 to 557) contain interpretation of the TGA results in relation to the quality of the pellets. As a reminder, our study employed various state-of-the-art analytical instruments including TGA for the characterization of pure and blended pellet samples made from Norway spruce and pea starch with intent to investigate differences in the mechanism of bonding relevant to the quality of the pellets. Standard practice for reporting investigations of this sort requires that conclusions been drawn on what information can be satisfactorily obtained from each instrument used in the analysis, which must be presented as a paragraph, after the presentation of the main results obtained from the instrument(s). Following this, a summary of what was investigated from the entire study, its main results and conclusions can then be presented in a separate section that is most suitable under the ‘’discussion and conclusions’’ section, which is what this manuscript has presented.

With regards to the evidence that justifies the statement in the lines in question (lines 538 to 557), data from the TGA plot provided the evidence from the degradation temperatures of the pellets and supported by credible literature citations (Refs 56 and 57). Recall that the quality of the pellets was determined in terms of burning rate and combustion efficiency (line 538), which are thermal parameters that can only be established through thermal analysis. The higher modification temperature for the 100% PSP in comparison to 100% NSP and the 50/50 blend is an indication that the former required more temperature to ignite than the latter two samples. This was an event practically experienced during TGA analysis of the pellets. Lower degradation/modification temperature means faster burning rate and better combustion efficiency due to greater contact between a material and its thermal oxidizing environment as well as due to the heating value of the material, which is often dependent upon chemical composition, particularly the concentration of carbon. The modification range temperature of the pellets has been made clearer in the plot in Figure 3.

For a convincing proof however, we challenge the reviewer to carefully perform a home-made experiment on the burning rate of any wood material and another material containing a significant amount of starch, just by ordinary ignition, in order to determine which burns faster with better combustion efficiency.

Point 13: Figures 2 and 3 appear to be rather essential contents of the paper. The contents of Fig. 2 being contradictory to the related text, and the contents of Fig. 3, along with the related text, being suspected by this reviewer to be incorrect reporting of measurements, this reviewer does not think he would have much of a possibility of understanding further discussion in the paper. For now, the review terminates at line 513.

Response 13: We have tried to be as comprehensive and clear as possible in the interpretation of the data in Figures 2 and 3. For better understanding, we encourage the reviewer to peruse through the following articles:

Anukam, A.I.; Berghel, J.; Famewo, E.B.; Frodeson, S. Improving the understanding of the bonding mechanism of primary components in biomass pellets through the use of advanced analytical instruments. J. Wood Chem. Technol. 2019, 0, 1–18. Gil, M.V.; Oulego, P.; Casal, M.D.; Pevida, C.; Pis, J.J.; Rubiera, F. Mechanical durability and combustion characteristics of pellets from biomass blends. Bioresour Technol. 2010, 101, 8859–8867. Zafar S. Summary of biomass combustion technologies. Available online: https://www.bioenergyconsult.com/tag/biomass-combustion-process/ (accessed 5 September 2019).

Reviewer 3 Report

The research work done by Anthony Ike Anukam research group “Characterization of pure and blended pellets made from Norway spruce and pea starch: A comparative study of bonding mechanism relevant to quality” is a good research work to study the pure and blended pellets. Authors documented a reasonable study of bonding mechanism relevant to quality which is very important for the current research and development towards to study further applications. The major advantage of the present documented report is “polar functional groups played a important role to understand the quality of the pellets based on bond strengths and the type of attraction forces between combining particles” which is pretty noteworthy demonstration in this research work. There are not many literature procedures, similar to present documented report in the literature for the pellets characterization. Given the importance of practicality for this work, I recommend the publication of this manuscript in the Energies.

Author Response

Comments and Suggestions for Authors

Point 1: The research work done by Anthony Ike Anukam research group “Characterization of pure and blended pellets made from Norway spruce and pea starch: A comparative study of bonding mechanism relevant to quality” is a good research work to study the pure and blended pellets. Authors documented a reasonable study of bonding mechanism relevant to quality which is very important for the current research and development towards to study further applications. The major advantage of the present documented report is “polar functional groups played a important role to understand the quality of the pellets based on bond strengths and the type of attraction forces between combining particles” which is pretty noteworthy demonstration in this research work. There are not many literature procedures, similar to present documented report in the literature for the pellets characterization. Given the importance of practicality for this work, I recommend the publication of this manuscript in the Energies.

Response 1: We agree with the reviewer that this is a good research work because studies involving differences in particle bonding mechanism between pure and blended biomass pellets is sparsely available in the literature. We also agree with the reviewer that our study is very important for the current research and development towards investigating further applications and we very much appreciate the reviewer’s recommendation of the publication of our manuscript.  

Reviewer 4 Report

Review comments on “Characterization of pure and blended pellets made from Norway spruce and pea starch: A comparative study of bonding mechanism relevant to quality” by Anukam et al.

The paper is very interesting. The methodology is coherent with the aim of the research and the proposed model is enough innovative.
My main comments are as below:

GENERAL COMMENTS:
- All of the abbreviations should be defined in full prior to it use. I suggest to add a specific table with all abbreviations.

ABSTRACT:
- Give in the last sentences of the abstract the main results of your work.

KEYWORDS:
- Do not repeat words of phrases from the title of the manuscript. I suggest to change "bonding".

Line 204-205: reduce line spacing.

Author Response

Comments and Suggestions for Authors

Review comments on “Characterization of pure and blended pellets made from Norway spruce and pea starch: A comparative study of bonding mechanism relevant to quality” by Anukam et al.

The paper is very interesting. The methodology is coherent with the aim of the research and the proposed model is enough innovative.

My main comments are as below:

Point 1: GENERAL COMMENTS:

- All of the abbreviations should be defined in full prior to it use. I suggest to add a specific table with all abbreviations.

Response 1: All abbreviations have been defined in full. It was not clear if adding a table with all abbreviations is in line with the journal’s requirements; however, a table that defines all abbreviations have been added in the first page of the manuscript.

Point 2: ABSTRACT:

- Give in the last sentences of the abstract the main results of your work.

Response 2: The abstract has been re-written and re-worded in a way that it now reflects the main results of the work.   

Point 3: KEYWORDS:

- Do not repeat words of phrases from the title of the manuscript. I suggest to change "bonding".

Response 3: This error has been noted and corrected. The word ‘’bonding’’ has now been replaced with ‘’biomass’’.   

Point 4: Line 204-205: reduce line spacing.

Response 4: The line spacing in lines 204-205 (now lines 226-227) has been reduced.   

Round 2

Reviewer 2 Report

It appears the Authors have not understood my remark regarding Fig. 2. Fig. 2 does not appear to contain significant absorbtion peaks. It does appear to contain signifixcant absorption gorges. The absorbtion gorges are most pronounced in the case of PSP (100%). The Figure reports greatest absorbance for the composite product, and lowest absorbance for the PSP.

The text does not refer to abosorbtion gorges but to absorption peaks. The text is in disagreement with the Figure. I suspect the Figure is incorrect.

It also appers that the Authors have neither understood my comment related to Fig. 3. Common sense proposes the Figure cannot be correct. Common sense proposes degradation activity increases with increasing temperature, whereas the Figure reports the greatest mass loss at the lowest temperature. The vertical axis possibly should be reversed, or the axis title changed.

Author Response

Point-by-point response to the reviewer's comments

Point 1: It appears the Authors have not understood my remark regarding Fig. 2. Fig. 2 does not appear to contain significant absorption peaks. It does appear to contain significant absorption gorges. The absorption gorges are most pronounced in the case of PSP (100%). The Figure reports greatest absorbance for the composite product, and lowest absorbance for the PSP.

Response 1: After a careful perusal of the figure in question (Fig. 2), the remark of the reviewer was clearly comprehended. The axis labelling have been changed from ‘’Absorbance’’ to ‘’Transmittance’’ and the related text completely re-written without contradictions. To avoid any ambiguity however, the peak numbering have been reduced in such a way that only the most important peaks relevant to chemical alterations and the type of attraction forces between particles of the pellets as well as their quality have been numbered. These have been made clear in the related text.

Point 2: The text does not refer to absorption gorges but to absorption peaks. The text is in disagreement with the Figure. I suspect the Figure is incorrect.

Response 2: As stated in the previous response, the text linked to the figure (Fig. 2) has been rephrased to reflect the interesting behavior of the composite material using the pure materials as reference points.

Point 3: It also appears that the Authors have neither understood my comment related to Fig. 3. Common sense proposes the Figure cannot be correct. Common sense proposes degradation activity increases with increasing temperature, whereas the Figure reports the greatest mass loss at the lowest temperature. The vertical axis possibly should be reversed, or the axis title changed.

Response 3: The reviewer’s comments about Fig. 3 were well comprehended and we agree with his/her remark that degradation activity increases with increasing temperature. This has been revised accordingly and the axis title of the figure in question changed. However, it is worth mentioning that unexpected behavior are sometimes experienced when materials undergo thermal analysis and that these could be attributed to a number of reasons including the type of material under analysis, its composition and thermal stability as well as analysis conditions.

Round 3

Reviewer 2 Report

Thank you.